# The Demographic Variation in Nutrition Knowledge and Relationship with Eating Attitudes among Chinese University Students

**DOI:** 10.3390/ijerph21020159

**Published:** 2024-01-31

**Authors:** Wen-Jing Deng, Ziyue Yi, John Chi-Kin Lee

**Affiliations:** 1Department of Science and Environmental Studies, The Education University of Hong Kong, Hong Kong, China; 2Academy of Applied Policy Studies and Education Futures, The Education University of Hong Kong, Hong Kong, China; jcklee@eduhk.hk

**Keywords:** nutrition knowledge, machine learning, eating attitudes, university students, China

## Abstract

There is a noticeable absence of health education among college students. This study aimed to evaluate the extent of general nutrition knowledge among Chinese university students and explore its association with eating attitudes. Data were collected from a group of 273 students in Spring of 2023, using a valid and reliable research instrument consisting of three sections: demographic variables, the General Nutrition Knowledge Questionnaire (GNKQ), and the Eating Attitudes Test (EAT-26). The results were analyzed using SPSS, with correlations and *t*-tests to examine the relationships between nutritional knowledge and dietary attitudes. Furthermore, the present study employed the random forest (RF) algorithm, a machine learning technique, utilizing the Mean Decrease Impurity (MDI) method to investigate the influence of various features on participants’ eating attitudes. The findings revealed that Chinese university students had an average accuracy of over 60% in their nutritional knowledge, but their understanding of the relationship between diet and disease still needs improvement. Moreover, male students had significantly lower nutritional knowledge than female students, and there was a positive correlation between nutritional knowledge and parents’ income. The study also found a significant correlation between the level of nutritional knowledge and eating attitudes. RF results indicated that the family income level exhibited the most substantial impact on the eating attitudes of the participants. The study highlights the need for nutrition education curriculum developers to focus more on improving students’ nutritional knowledge, with particular attention given to male students, low-income individuals, and those with an abnormal BMI.

## 1. Introduction

The nation’s health has always been a national priority, and poor lifestyle habits and diet structure have contributed to many health problems in recent years. University students, in particular, are in the middle stages of growth and development and have a high demand for various nutrients [1]. University life represents a significant lifestyle change, as students often live independently for the first time and must take on the responsibility of buying and preparing their meals and managing a new meal plan [2]. 

Several previous studies have shown that college students lack sufficient nutrition knowledge to make informed dietary decisions [2,3]. Nutrition knowledge has been extensively studied in various countries. For instance, a cross-sectional survey conducted in Poland found that nutrition education programs that focus solely on factual knowledge may not effectively promote appropriate behaviors in students, as they may neglect the importance of positive attitudes toward food and nutrition [4]. The significance of nutritional knowledge is further highlighted by the challenges of obesity in Kenya, South Africa (SA), and the United Kingdom (UK) [5,6]. Moreover, an Australian study utilized the knowledge–attitude–behavior model to suggest that strategies aimed at overcoming cognitive barriers and enhancing self-efficacy are necessary to encourage positive eating behaviors [7].

Regrettably, as of 2020, 30% of Chinese university students are failing in physical fitness, and some researchers posit that this shortfall may be due to a lack of nutritional knowledge [2,3,8,9]. University students, who are particularly susceptible to societal pressures to maintain a thin physique [2,3], may suffer from this knowledge gap. Nutritional knowledge, as defined by the Food and Agriculture Organization (FAO), refers to an individual’s understanding of nutrition. This understanding encapsulates an individual’s cognitive capacity to recall and comprehend food- and nutrition-related terminology, specific details, and factual information [10]. Foundational knowledge of nutrition is crucial for understanding the delicate balance between energy input (calories in food and drink, etc.) and capacity expenditure (basal metabolism, exercise, etc.), which plays a significant role in preventing obesity and non-communicable diseases [11].

Unhealthy diets contribute to global health risks [12], and long-term nutritional deficiencies among university students can lead to chronic diseases such as obesity, cardiovascular disease, and even certain types of cancer at an early age [13]. Eating disorders have also become prevalent among university students [14]. Obesity is defined using the body mass index (BMI), with a BMI of 25 to <30 considered overweight and a BMI of 30 or above classified as obese [15]. In 2016, almost one-third of China’s children and adolescents were classified as overweight or obese [16]. Eating disorders are marked by significant disruptions in eating patterns and body weight regulation [17]. Numerous adolescents in Hong Kong face considerable dissatisfaction with their body weight. Teenage girls, individuals who are overweight, and those with poor academic performance are at a higher risk of developing disordered eating [18]. They are prevalent among young adults and are increasingly becoming more common in China [19,20,21].

The objective of this study is to comprehensively examine the level of general nutritional literacy among a sample of Chinese university students using a validated and reliable instrument and to analyze variations in knowledge levels based on demographic factors. Furthermore, despite voluminous literature on nutrition knowledge, no study has investigated the association between eating attitudes and nutritional knowledge among college students in China. This study aims to investigate potential links between nutritional knowledge and eating attitudes among a group of university students. It is hoped that this study will provide a reference for improving nutrition education, enhancing nutrition knowledge, and correcting eating attitudes and behaviors, and serving as a reference for research on the nutritional diet of students in China.

## 2. Methods

### 2.1. Questionnaire Selection and Translation

Data were collected on various factors, including age, sex, height, weight, hometown region, academic qualifications, family income level, body satisfaction, and attendance at nutrition courses from January to May 2023. The BMI diagnostic criteria used for the study were based on the World Health Organization (WHO) BMI criteria for the Asian population [16]. Family income levels were divided into four categories, and hometown regions were divided into seven administrative and geographical divisions of China. Body satisfaction was measured on a 5-point scale.

The General Nutrition Knowledge Questionnaire (GNKQ) has been extensively used to evaluate nutrition knowledge and has demonstrated strong reliability and validity when applied to individuals in the UK [22]. The Chinese version of GNKQ was primarily adapted from the Turkish version [23] and was modified to include Chinese foods that align with the Chinese Dietary Guidelines [24]. Therefore, this study questionnaire is based on the GNKQ initially designed by Parmenter and Wardle [22] and the C-GNKQ suitable for China [23], with changes made to the spelling and wording of some questions. The content of the questionnaire contains both the breadth of questions from the original GNKQ and Chinese food from the C-GNKQ. Questions that duplicated the content of the examination were removed, taking into account the participants’ patience in completing the questionnaire.

The EAT-26 is a widely accepted and reliable instrument for assessing eating attitudes. Garner et al. [25] proposed a simplified version based on a factor analysis of the original scale (EAT-40), comprising 26 items presented in a self-report format. The scale is highly correlated with the original EAT-40 and has been used to measure eating disorder symptoms. The Chinese version of the EAT-26 was used in this study, and scores were calculated by summing the composite items, with scores of 20 or more indicating disordered eating.

### 2.2. Participant and Sample Size

To recruit the target number of participants, the study was advertised via social media, advertising platforms, and assistance from friends and family. Once the students agreed, a link to the questionnaire was shared for completion. Eligibility for participation required students to be 18 years or older and currently enrolled in a university program, including Bachelor’s, Master’s, or PhD degrees. Prior to completing the questionnaire, participants signed a consent form, acknowledging that the data collected would not adversely affect them. Of the returned 336 questionnaires, 273 were considered valid. Quota sampling was employed to recruit these participants, ensuring the sample mirrored the general population. An analysis of the participants’ hometowns showed broad representation from all regions of China, aligning with each region’s population proportion (Figure 1).

The reliability of the Eating Attitudes Test (EAT-26) was measured using Cronbach’s alpha. The scale, divided into diet, bulimia and food preoccupation, and oral control dimensions, had a Cronbach’s alpha of 0.870, indicating high reliability. Table 1 provides reliability estimates for these dimensions, with diet at 0.811, bulimia and food preoccupation at 0.654, and oral control at 0.718. These results signify the questionnaire’s satisfactory reliability. Internal consistency, which measures the extent to which all items in a scale assess different aspects of the same attribute or construct, was also evaluated. The overall questionnaire had a Cronbach’s alpha of 0.847, demonstrating acceptable reliability. However, the dietary recommendations and food choices sections did not meet the reliability criteria due to the small number of items (Cronbach’s alpha < 0.7). On the other hand, the food sources and nutrients and diet–disease relationships sections both met the reliability criteria (Cronbach’s alpha > 0.7), indicating that the questionnaire’s overall reliability is satisfactory.

### 2.3. Statistical Analysis

#### 2.3.1. Quantitative Research

The questionnaire was administered and supervised only by the researcher and their supervisor. Descriptive statistics were used to analyze demographic variation, and the relationship between demographic variables and nutrition knowledge scores was examined using a bivariate correlation analysis and multiple linear regression analyses. Gender, income level, and weight status effects were compared using a two-sample *t*-test and ANOVA. The relationship between eating attitudes and nutritional knowledge was analyzed using a bivariate correlation analysis and a *t*-test. All statistical analyses in this study were conducted using SPSS v22.0 (IBM Corporation, Armonk, NY, USA) [26]. 

#### 2.3.2. Random Forest (RF) with Machine Learning (ML)

This study used a random forest algorithm to analyze how physical attributes and personal characteristics affect eating attitudes. Factors like height, weight, BMI, gender, age, mainland region of origin, and family income level were considered, assessing eating attitude using the EAT-26. The dataset consisted of 273 data samples, which was preprocessed and divided into two sets: the training set and the validation set. Python 3.9.12 was used to build the random forest model, with the parameter details shown in Table 2, using the Classification and Regression Tree (CART) algorithm to construct individual trees. The Gini Impurity was used to examine the impact of physical attributes and personal characteristics on eating attitudes. The training time was approximately 6 h, and 26,880 random forest models were established. The Accuracy Score was used to evaluate the performance and predict the prediction error of the models. The optimal model was found with a maximum Accuracy Score of 0.7804878048780488. The Mean Decrease in Impurity (MDI) was a metric used in the random forest model to assess the importance of input features in predicting a target variable. MDI calculated the average reduction in impurity achieved by splitting a feature across all trees. A higher MDI score indicated a more significant impact on impurity reduction, while a lower score indicated less influence.

## 3. Results

### 3.1. Demographic Characteristics of University Students

The study included 273 university students, with a geographically diverse sample from all regions of China (Table 3). The majority of participants were female (60.1%) and between the ages of 18 and 24 years (84.2%). Most participants held undergraduate degrees (72.8%) and fell within the RMB 10,000–50,000 annual family income bracket (68.9%). The analysis of BMI showed that an overweight status and obesity were more prevalent in males, while an underweight status was more prevalent in females. Overall, the study provided a representative sample of nutritional knowledge levels across China.

The mean correctness rate of the GNKQ score was 65.9 ± 12.9%. As shown in Table 4, the scores for knowledge of diet–disease relationships and choosing everyday foods tended to be lower than those for the other sections. Each sub-questionnaire had a score of 0, and all but the second part had a full score, indicating a wide range of participants’ nutritional knowledge levels.

Figure 2 presents the results of Pearson’s correlation analysis, revealing that females outperformed males in terms of nutritional knowledge (*p* < 0.01, r = 0.288), and higher income levels were associated with higher scores (*p* < 0.01, r = 0.337). On the other hand, academic qualifications (*p* < 0.01, r = −0.120) and participation in courses (*p* < 0.01, r = −0.134) were negatively correlated with nutritional knowledge, although these correlations were weak. Notably, the correlation between age, region, BMI, and overall performance did not reach a significant level. In terms of the sub-questionnaire, the results indicated that only gender and family income levels exhibited a significant correlation with each subscale.

To determine whether the observed relationships between variables remained significant after controlling for socio-demographic variables, a regression analysis was conducted. The model fit well (R2 = 0.231 > 0.2), indicating that the results can truly and reliably reflect the influence of gender, age, BMI, hometown region, academic qualifications, family income level, and participation in nutrition courses on nutrition knowledge scores. There was no collinearity between the seven variables, and all VIF values were less than 5. The Durbin–Watson statistic was 1.726, close to 2 (1.7–2.3), indicating no autocorrelation in the residual sequence. At least one of the seven independent variables could significantly influence the dependent variable (F = 11395, *p* < 0.001). According to the data, gender (B = 5.484, *p* < 0.001), family income level (B = 4.987, *p* < 0.001), and BMI (B = 3.341, *p* < 0.01) could significantly and positively predict nutrition knowledge scores. The final regression equation was as follows: nutritional knowledge = 23.449 + 5.484 × gender + 2.625 × income + 0.492 × BMI.

### 3.2. Difference in GNKQ Overall Scores

As shown in Figure 3, the study found that females had significantly higher levels of nutritional knowledge than males (*p* < 0.001). Additionally, a positive correlation existed between family income and nutrition knowledge levels, with higher-income groups demonstrating higher scores. Pairwise comparisons found that the overall knowledge score of the family income above RMB 50,000 was significantly higher than the other three groups (*p* < 0.001). However, no significant difference was found in nutrition knowledge scores between individuals with different BMI levels.

### 3.3. Nutrition Knowledge and Eating Attitudes

#### 3.3.1. Analysis of the Correlation between the Level of Nutritional Knowledge and Eating Attitudes

A Pearson correlation analysis was conducted on the level of nutritional knowledge and eating attitudes (Table 5). The results showed that there was a significant weak negative correlation between the overall nutrition knowledge score and eating attitudes (r = −0.128, *p* < 0.05). This means that the higher the level of nutritional knowledge, the lower the tendency to develop eating disorders. In the sub-questionnaire, part 1 and part 4 had a significant negative relationship with eating attitudes, while the remaining two parts did not show a significant correlation with eating attitudes.

Of the participants, 56 were thought to have an eating disorder (EAT-26 scores above 20). The results of the test showed a significant difference (*p* < 0.05), meaning that those who did not suffer from an eating disorder scored higher in nutritional knowledge.

#### 3.3.2. Machine Learning Analysis among the Impact of Different Features on Eating Attitudes

The present study used machine learning (ML) methods to compare participants’ eating attitudes as influenced by different features. Eight different features were used in the study, as shown in Figure 4. It shows the feature importance ranking based on the Mean Decrease Impurity (MDI) method, with the error bars indicating the standard deviation of the importance of individual trees in the random forest model. The findings show that the most important feature is family income level (FIL). Next, height (Height) and weight (Weight) are the second and third most important features influencing eating attitudes. Therefore, there is good reason to believe that the family income level had the most significant influence on participants’ eating attitudes.

## 4. Discussion

### 4.1. Demographic Characteristics of University Students

Previous research studies have demonstrated that college students do not possess sufficient nutrition knowledge to make informed dietary decisions [2,3]. In a study that discussed nutritional knowledge among college students, the average student nutrition knowledge score was 58% [27]. Research suggests that the basics of healthy eating are relatively easy for individuals to understand and grasp, but there is confusion about the deeper knowledge of nutrition. An Australian nutrition knowledge survey of the community population found that basic dietary guidelines had been disseminated to the community, but detailed knowledge of the nutritional content of foods and knowledge of food choices were scarce [28]. There is a lot of research on diet-related diseases, but little knowledge among the public [29].

It is worth noting that there were significant differences between males and females in BMI, and previous studies have supported the fact that males are more likely to be overweight [21,30]. Over the past few decades, the ideal body image portrayed by mass media has emphasized strong men and slim women [31]. Studies indicate that females are more prone to overestimating their weight, even when their actual weight falls within the healthy range. This tendency can contribute to the adoption of unhealthy behaviors aimed at weight control [21]. Consequently, the prevalence of overweight individuals is significantly lower among females compared to males, and vice versa.

### 4.2. Demographic Differences in Nutritional Knowledge

Gender differences in nutritional knowledge have been studied in several previous studies, with females demonstrating superior knowledge in all areas of nutrition [32]. Women are also more family-oriented, which makes the health aspect of the family more of a concern for them [29]. The results of this study showed that nutritional knowledge was positively correlated with the income level, which is consistent with the findings of a study in China by Gao et al. [23]. Low-income adult Americans often have diets that are high in sugar and fat and low in whole grains, vegetables, and fruit [33] However, it is important to note that the educational level has been found to be positively correlated with the income level [34]. This suggests that a lack of nutritional knowledge may contribute to the dietary patterns observed among low-income individuals. Considering the context of China, where people tend to consume more vegetables compared to their Western counterparts, the same conclusion can be made from this study. Various nutrition education interventions have been developed and implemented to increase nutrition knowledge and narrow the gap between low and high incomes, including community-based programs, school-based programs, and workplace wellness initiatives, among others.

BMI is an important variable in the study of nutritional knowledge, and many studies have demonstrated a significant correlation between nutrition knowledge and BMI levels [35,36]. However, the results of O’brien and Davies showed that there was no significant correlation between the level of nutritional knowledge and BMI, which is consistent with the results of this research [37]. A comparison of the levels of nutritional knowledge of obese and non-obese children suggests that the origin of childhood obesity is not solely related to a lack of nutrition knowledge [38]. While nutrition knowledge is an important component of healthy eating, it is not the only factor in determining an individual’s weight status, and interventions aimed at preventing an overweight status and obesity should take a comprehensive approach that addresses a wide range of factors.

### 4.3. Nutrition Knowledge and Eating Attitudes

Food-related cognitions are an important variable for binge eating [30]. The study made a surprising finding that 56 participants, accounting for 20.5% of the total participants, had eating disorders in this study, which is a significant and concerning result. A study conducted by Korinth et al. in German universities found that senior nutrition students made healthier food choices and were less inclined to be compulsive about their eating behavior [39]. Some studies have also indicated that there is no significant difference in nutritional knowledge between eating-disordered and non-eating-disordered adolescents and their parents, but both suffer from a lack of basic nutritional knowledge [40]. Therefore, providing more knowledgeable information may reduce the risk of eating disorders.

Nutritional knowledge shapes eating attitudes, leading to a varied diet and avoiding unhealthy food choices. Educational interventions in a supportive school environment can promote healthy eating habits and alleviate the issue of extreme weight control behaviors.

### 4.4. Implications and Conclusions

By examining the general level of nutritional knowledge and demographic differences in knowledge levels, this study provides important insights into the factors that may influence dietary behaviors in this population. Nutrition education curriculum developers should focus more on improving students’ nutritional knowledge, with particular attention given to males, low-income people, and those with abnormal BMI. Schools must prioritize these groups to ensure that they receive adequate nutrition education and support for healthy eating habits. Additionally, this dissertation fills a research gap in the relationship between dietary attitudes and nutritional knowledge among university students from a nutrition education perspective in China. Our findings provide a reference for research on the nutritional diets of Chinese university students, especially given the greater context of the obesity crisis.

In conclusion, this study underscores the importance of nutritional knowledge in promoting healthy dietary behaviors and preventing obesity and eating disorders. The findings suggest that interventions aimed at improving nutritional knowledge together with healthy exercise habits could be targeted, taking into account the demographic differences in knowledge levels. Such interventions should be implemented in a supportive environment, such as schools and universities, and should focus on providing adequate information and support to those who need it most. By doing so, we can promote healthy eating habits and prevent the negative health consequences associated with poor dietary choices.

### 4.5. Limitations and Suggestions for Future Research

This study’s limitations primarily lie in the scope of the study, the sample size, and the sample’s representativeness. In terms of scope, the research evaluated the connection between nutritional knowledge and eating attitudes among university students in China, focusing on differences in nutritional knowledge levels based on gender, family income, and BMI. However, nutritional knowledge is influenced by more than just demographic factors. It can also be affected by aspects such as profession, parental education levels, academic stress, personal interests, and attitudes. Due to the anonymous and online nature of the responses, the validity of the questionnaire responses could be questionable, presenting another limitation. There is no assurance that participants were free from external disturbances while answering the questionnaire or that they did not use outside resources for assistance. Lastly, the unequal gender representation among the participants could have potentially influenced the study’s results. 

## Figures and Tables

**Figure 1 ijerph-21-00159-f001:**
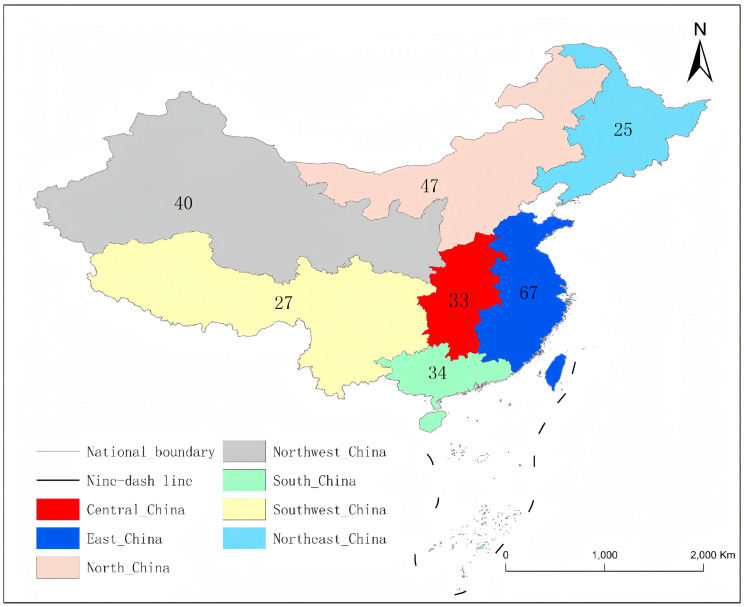
National distribution of participants. *Note.* The numbers on the map refer to the participants in each area.

**Figure 2 ijerph-21-00159-f002:**
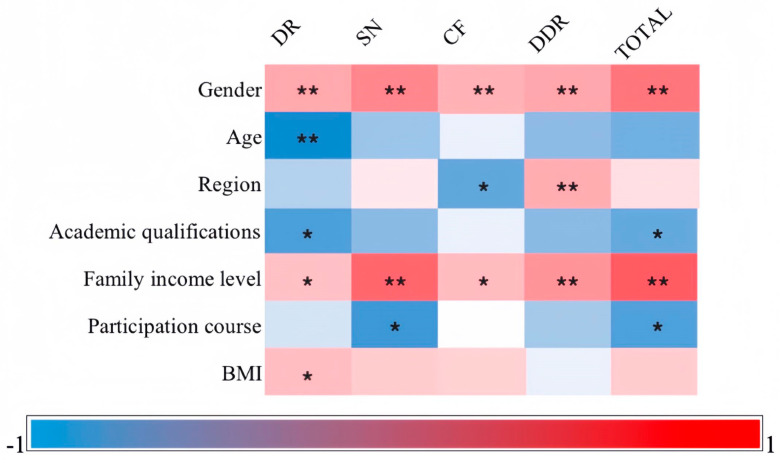
Factors related to nutritional knowledge. *Note.* ** *p* < 0.01, * *p* < 0.05. DR: dietary recommendations, SN: food sources and nutrients, CF: choosing everyday foods, and DDR: diet–disease relationships.

**Figure 3 ijerph-21-00159-f003:**
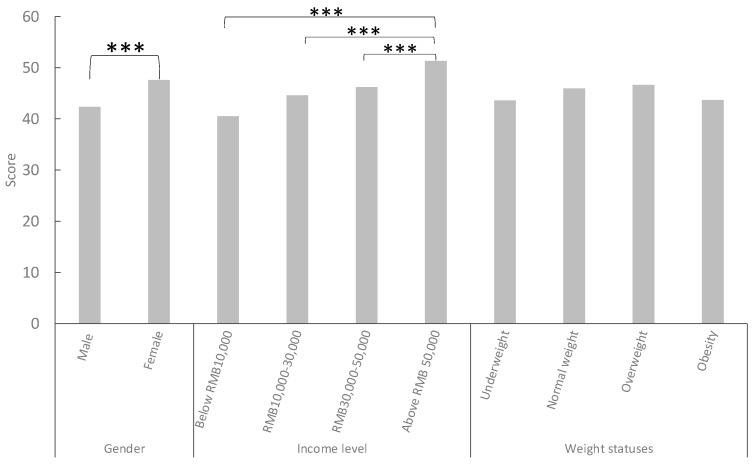
Comparison of GNKQ overall scores between different genders, income levels, and weight statuses. *Note.* *** *p* < 0.001.

**Figure 4 ijerph-21-00159-f004:**
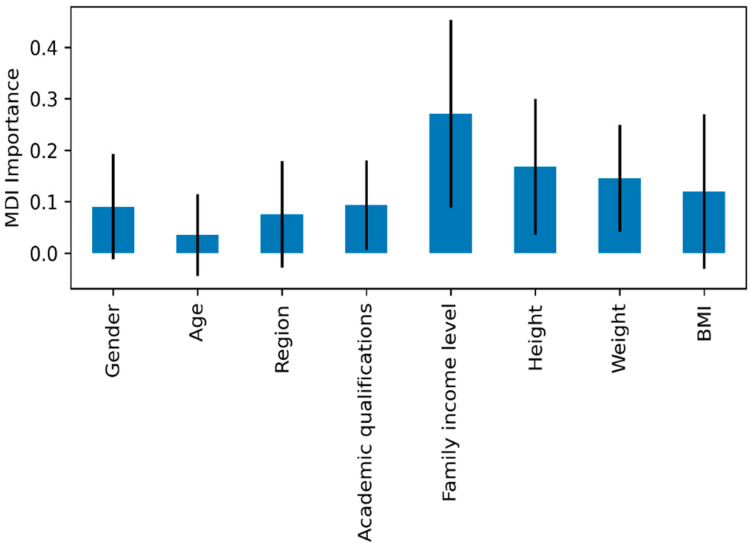
Random forest feature importance based on Mean Decrease in Impurity (MDI) among eating attitudes.

**Table 1 ijerph-21-00159-t001:** Reliability Analyses (Cronbach’s Alpha, α) of scale.

	Dimensions	Reliability (Cronbach’s a) (*n* = 273)	Number of Items
EAT-26	Diet	0.814	13
	Bulimia and food preoccupation	0.754	6
	Oral control	0.677	7
	Total	0.87	26
GNKQ	Dietary recommendations	0.364	9
	Food sources and nutrients	0.818	43
	Choosing everyday foods	0.102	4
	Diet–disease relationships	0.752	13
	Total	0.847	69

**Table 2 ijerph-21-00159-t002:** Parameter Details of Selected Random Forest Model.

Parameter	Value
Number of Trees	10
Maximum Depth of Tree	NULL
Information Gain Function	Gini Impurity
Method of Finding Number of Features When Looking for The Best Split	Square Root
Minimum Number of Samples at One Leaf Node	8
Minimum Number of Samples for Splitting an Internal Node	9

**Table 3 ijerph-21-00159-t003:** Demographic Characteristics of University Students Participating in the Study.

Characteristics	Overall (*n* = 273)		Male (*n* = 109)		Female (*n* = 164)	
	*n*	%	*n*	%	*n*	%
Age						
18–24 y	230	84.2	84	77.1	136	82.9
25–34 y	33	12.1	14	12.8	19	11.6
35 y and over	10	3.7	2	1.8	8	4.9
Region						
Northwest China	40	14.7	10	9.2	30	18.3
Northeast China	25	9.2	9	8.3	16	9.8
North China	47	17.2	21	19.3	26	15.9
South China	34	12.5	16	14.7	18	11
East China	67	24.5	30	27.5	37	22.6
Southwest China	27	9.9	10	9.2	17	10.4
Central China	33	12.1	13	11.9	20	12.2
Academic qualifications						
Bachelor’s	196	71.8	81	74.3	115	70.1
Master’s	64	23.4	20	18.3	44	26.8
PhD	13	4.8	8	7.3	5	3
Family income level						
Below RMB 10,000	44	16.1	23	21.1	21	12.8
RMB 10,000–30,000	96	35.2	40	36.7	56	34.1
RMB 30,000–50,000	92	33.7	35	32.1	57	34.8
Above RMB 50,000	41	15	11	10.1	30	18.3
BMI						
Underweight	49	17.9	13	16.5	36	25.6
Normal weight	173	63.3	61	57.8	112	65.9
Overweight	44	16.1	30	23.9	14	7.9
Obesity	7	2.6	5	1.8	2	0.6

**Table 4 ijerph-21-00159-t004:** Mean and Range of Correct Scores of GNKQ.

Knowledge Components (No. of Items)	Min	Max	Mean (%)	SD
Dietary recommendations (9)	0	9	7.43 (82.56%)	1.21
Food sources and nutrients (43)	0	39	27.72 (71.08%)	6.59
Choosing everyday foods (4)	0	4	2.45 (61.25%)	0.95
Diet–disease relationships (13)	0	13	7.89 (60.69%)	2.8
Overall knowledge score (69)	12	60	45.49 (65.9%)	8.9

**Table 5 ijerph-21-00159-t005:** Correlation Analysis between Nutritional Knowledge and Eating Attitudes and Differences in the GNKQ on Eating Disorders vs. Non-eating Disorders.

Variables	Overall (*n* = 273) (Mean ± SD)	EAT-26		r	T
		≥20 (*n* = 56) (Mean ± SD)	<20 (*n* = 217) (Mean ± SD)		
Dietary recommendations	7.43 ± 1.208	7.04 ± 1.651	7.53 ± 1.045	−0.273 ***	−2.152 *
Food sources and nutrients	27.72 ± 6.593	26.48 ± 8.122	28.04 ± 6.119	−0.06	−1.342
Choosing everyday foods	2.45 ± 0.954	2.38 ± 1.071	2.47 ± 0.923	−0.023	−0.632
Diet–disease relationships	7.89 ± 2.799	6.91 ± 3.282	8.14 ± 2.61	−0.153 *	−2.967 **
Overall knowledge score	45.49 ± 8.898	42.80 ± 10.87	46.18 ± 8.202	−0.128 *	−2.170 *

*Note.* r = relationship between nutritional knowledge and eating attitudes; T = significance level of the difference between the mean with and without eating disorders; *** *p* < 0.001, ** *p* < 0.01, * *p* < 0.05.

## Data Availability

The data presented in this study are available on request from the corresponding author. The data are not publicly available due to privacy of test data.

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
