# Peer review of "The Demographic Variation in Nutrition Knowledge and Relationship with Eating Attitudes among Chinese University Students"

_ijerph, 2024, doi:10.3390/ijerph21020159_

Round 1

Reviewer 1 Report

Comments and Suggestions for Authors

The manuscript entitled "Demographic Variation in Nutritional Knowledge and Relationship with Eating Attitudes among Chinese University Students” is the study aiming to determine the general nutritional knowledge among Chinese university students and its variations according to demographic factors. This study also aims to investigate the relationship between nutritional knowledge and eating attitudes among university students

The manuscript is nicely and clearly written. It deals with the important and always interesting topic of the influence of health and habits on people's health

The biggest drawback, as the authors themselves say, is the small number of respondents. Given the way in which the study was conducted (online survey), the question arises as to whether more effort could have been made? Also, was the power of the study calculated?

It is not clearly stated whether you checked whether all participants were of the same nationality or whether foreign students/guest students at the same university could also be included in the study?

Comments on the Quality of English Language

The English language is satisfactory, minor corrections relating more to technical matters, such as double commas, double spaces and minor language errors

Author Response

Dear Reviewer 1, 

As the corresponding author, I would like to express my gratitude for the reviewer's valuable feedback. We have carefully revised the manuscript based on the reviewers' comments, and we believe that the revisions have significantly improved the quality and clarity of our work. The detailed responses to each comment can be found in the table provided in the letter.

Should you require any further information or have any additional inquiries, please do not hesitate to contact me.

Sincerely,

Dr. WJ Deng

Reviewer 2 Report

Comments and Suggestions for Authors

The authors reported that the objective of this study is to examine the level of general nutritional literacy among a sample of Chinese university students, and to analyze variations in knowledge levels based on demographic factors. In other words, an analytical as well as descriptive purpose has been set forth. This goal can be achieved with a cross-sectional study. However, the authors did not mention the study type, nor did they explain how they calculated the sample size and how decided that they had reached a sufficient number of samples. In lines 159th and 160th, it was reported that overall, the study provided a representative sample of nutritional knowledge levels across China. It is far from scientific accuracy to declare that a total of 273 students can represent all Chinese university students. Because only volunteers participated, participants were probably determined by the snowball method, and it is also unclear how many university students there are in the mentioned seven regions and whether a proportional sample was taken. The study design, sample size and sample selection methods should be defined in more detail in the method section, not in the limitations.

The authors stated that academic competencies and participation in courses are negatively associated with nutritional knowledge. This finding, which is not compatible with the literature knowledge, was not commented on in the discussion section. In particular, the fact that the references used in the discussion section date back 10 years and earlier raises concerns that the results are not discussed correctly and adequately. According to the journal rules, references should be numbered. However, APA 6th format was used in lines 61, 66, 67, 71, 90, 97, 98, 105, 254 and 262.

Author Response

Dear Reviewer 2, 

As the corresponding author, I would like to express my gratitude for the reviewer's valuable feedback. We have carefully revised the manuscript based on the reviewers' comments, and we believe that the revisions have significantly improved the quality and clarity of our work. The detailed responses to each comment can be found in the table provided in the letter.

Should you require any further information or have any additional inquiries, please do not hesitate to contact me.

Sincerely,

Dr. WJ Deng
